# OX-HDL: A Starring Role in Cardiorenal Syndrome and the Effects of Heme Oxygenase-1 Intervention

**DOI:** 10.3390/diagnostics10110976

**Published:** 2020-11-20

**Authors:** Stephen J. Peterson, Abu Choudhary, Amardeep K. Kalsi, Shuyang Zhao, Ragin Alex, Nader G. Abraham

**Affiliations:** 1Department of Medicine, Weill Cornell Medicine, New York, NY 10065, USA; Stp9039@nyp.org; 2Department of Medicine, New York Presbyterian Brooklyn Methodist Hospital, Brooklyn, NY 11215, USA; Abc9031@nyp.org (A.C.); akk9013@nyp.org (A.K.K.); shz9027@nyp.org (S.Z.); 3Department of Medicine, New York Medical College, Valhalla, NY 10595, USA; ralex@nymc.edu; 4Department of Pharmacology, New York Medical College, Valhalla, NY 10595, USA; 5Department of Medicine, Joan C. Edwards School of Medicine, Marshall University, Huntington, WV 25701, USA

**Keywords:** heme oxygenase, cardiorenal syndrome, renal failure, congestive heart failure, oxidized HDL, HDL proteome

## Abstract

In this review, we will evaluate how high-density lipoprotein (HDL) and the reverse cholesterol transport (RCT) pathway are critical for proper cardiovascular–renal physiology. We will begin by reviewing the basic concepts of HDL cholesterol synthesis and pathway regulation, followed by cardiorenal syndrome (CRS) pathophysiology. After explaining how the HDL and RCT pathways become dysfunctional through oxidative processes, we will elaborate on the potential role of HDL dysfunction in CRS. We will then present findings on how HDL function and the inducible antioxidant gene heme oxygenase-1 (HO-1) are interconnected and how induction of HO-1 is protective against HDL dysfunction and important for the proper functioning of the cardiovascular–renal system. This will substantiate the proposal of HO-1 as a novel therapeutic target to prevent HDL dysfunction and, consequently, cardiovascular disease, renal dysfunction, and the onset of CRS.

## 1. Introduction

Admission of heart failure (HF) patients in American hospitals exceeds 1 million people annually [1], with heart failure or coronary artery disease more prevalent in the more advanced stages of chronic kidney disease (CKD) [2]. Patients with heart failure are more predisposed to the development of acute kidney injury (AKI) [3,4]. This sets the stage for the relevance of the cardiovascular–renal system as an interrelationship between kidney function and cardiovascular health. Epidemiologic studies have investigated how renal dysfunction is a prominent risk factor for cardiovascular disease (CVD) [5] and how heart failure predisposes to kidney damage and/or the exacerbation of chronic kidney dysfunction. [6,7] However, in vitro and in vivo studies have yet to establish a definitive pathophysiological explanation for this phenomenon [8], otherwise known as cardiorenal syndrome (CRS) [9].

High-density lipoprotein–cholesterol (HDL–C) levels have become promising markers for the risk of CVD and even CKD [10,11]. Recent studies have cited that HDL function is more important than levels and that remodeling and dysfunction likely contribute to increased risk of CVD, CKD, and CRS [12,13,14,15,16]. HDL utilizes its protective effect through multiple mechanisms, including lowering tissue cholesterol levels through reverse cholesterol transport, attenuation of low-density lipoprotein (LDL) oxidation, and decreasing inflammatory responses via association with paraoxonase 1 (PON1) [17,18,19,20,21,22,23].

Dysfunctional HDL can result from free radical attack or oxidation of “good” HDL, leading to Ox-HDL (“bad” HDL) [24,25,26]. Lipids and lipoproteins are the major culprits of free radical damage [27], which results in lipid peroxidation. Free-radical-mediated lipid peroxidation alters the biophysical properties of cell membranes, which may impair normal cellular function [28]. Furthermore, the generation of lipid peroxidation products, i.e., F2-isoprostane from arachidonic acid, may propagate the free radical damage via covalent modification of biomolecules [29,30,31]. It is pertinent to find an endogenous antioxidant that can prevent the remodeling of “good” HDL to proinflammatory and atherogenic “bad” HDL [32,33,34,35] (Figure 1). For CRS, it is crucial to find an antioxidant that can directly target the cardiovascular–renal system to protect from oxidative damage and dysfunction to the cardiovascular or renal systems [36].

## 2. The Heme Oxygenase System

Heme oxygenase-1 (HO-1) is an inducible enzyme within the body that is responsible for the catabolism of heme to equimolar parts carbon monoxide (CO) and biliverdin/bilirubin and the release of free iron [37]. HO-1 is known as a stress response protein [38], and we associate its induction with protection against reactive oxygen species (ROS) and, subsequently, oxidative stress [39]. We find heme oxygenase in the kidney in two isoforms: HO-1 (inducible form) and HO-2 [40]. Studies have shown HO-1 to be necessary and important for renal vascular and tubular function [41]. HO-1 has been shown to be important in vascular protection and function through the induction of adiponectin [42,43,44]. HO-1 is a potential and relevant therapeutic target for protection against and amelioration of CRS. HO-1 has been shown to decrease levels of angiotensin-II-mediated isoprostane production in endothelial cells [45]. Heme oxygenase has been proven to be cardioprotective, with associated induction of adiponectin expression [43,44,46,47], reduction in Ox-HDL with suppression of isoprostane production, and isoprostane binding to HDL. [48] Additionally, HO-1 may protect against heme-mediated damage in CRS, as hemoglobin has been shown to bind to “good” HDL and alter its conformation and functionality, causing HDL to become proinflammatory and atherogenic. [41,49].

Adiponectin is a cardioprotective protein hormone and has been shown to have increased plasma levels and improve vascular function after induction of HO-1 [37]. Adiponectin is well established as necessary for proper cardiovascular health, with clinical investigations proving adiponectin deficiency (hypoadiponectinemia) as an independent risk factor for CVD [50]. Researchers have also found hypoadiponectinemia to be strongly associated with renal dysfunction and CKD [51,52,53], and circulating adiponectin levels may be a predictor for CKD [54]. These studies show that adiponectin is crucial for vascular and renal function, with decreased levels leading to cardiovascular–renal dysfunction.

The heme oxygenase system may act as a potential therapeutic target for protection against and amelioration of cardiorenal syndrome through many pathways. Heme oxygenase is a potent endogenous antioxidant [55], and pharmacological induction of HO-1 improved type-1 cardiorenal syndrome in postischemic SCID mice [56]. HO-1 has been shown to decrease levels of angiotensin-II-mediated isoprostane-induced oxidative stress production in endothelial cells [45].

## 3. Structure of HDL and Reverse Cholesterol Transport Pathway

HDL is a small, dense lipoprotein particle with a high ratio of proteins to lipids [57], synthesized by liver hepatocytes [58]. HDL is a powerful anti-inflammatory agent that inhibits atherogenesis [59,60]. The HDL proteome is very complex, with over 550 proteins reported in HDL [9]. HDL proteomics is a relatively novel approach to understanding the complex makeup and function of HDL in the setting of oxidative stress [61,62]. Sixteen HDL species with distinct proteomic signatures have been identified [63]. The HDL proteome is complex and separate from HDL cholesterol. Mass spectrometry has been used to understand complete HDL analysis and to identify biomarkers in order to better understand HDL function [61,64,65,66,67]. In fact, the HDL lipoprotein proteome has a high correlation with risk factors for cardiovascular disease and atherosclerotic burden and calcification on CT angiograms of coronary arteries [68]. The diversity of the HDL proteome is associated with clinical outcomes in patients with heart failure [69]. Alterations in the HDL proteome have been shown to result in dysfunctional HDL particles in type I diabetics [70,71].

HDL is primarily composed of several apolipoproteins, including Apo A-1, Apo A-II, and other proteins, including the enzyme paraoxonase. The external layer of HDL comprises free cholesterol, apolipoproteins (e.g., apoA-I, apoA-II, apoC, apoE), and phospholipids and is amphipathic [72]. The inner core of the HDL particle is highly concentrated with cholesterol esters, contains a minute amount of triglycerides, and is, therefore, hydrophobic [73].

HDL-C is considered the “good cholesterol” because of the physiologic function it performs in “reverse cholesterol transport” [9,74]. This is the process where the HDL particles move through the circulation and extract free cholesterol from less-dense particles and transport the free cholesterol to the liver to be processed and expelled, reducing the overall level of total cholesterol [75]. HDL is the smallest of the lipoproteins but contains the highest apolipoprotein/ lipid ratio [67]. The cholesterol delivered to the liver is excreted in bile and eventually converted into bile acids [76]. Delivery of HDL cholesterol to organs like adrenals, ovaries, and testes is critical for steroid hormone synthesis [34,77]. The efflux of cholesterol from HDL involves the following regulatory proteins.

ApoA-I has been shown to bind cholesterol in vitro and in vivo, showing its role in the uptake of extrahepatic cholesterol by HDL while traveling through the circulation [78,79,80]. Infusing patients with pro-apo-A-I was shown to increase RTC [81,82]. ApoA-I is synthesized in and interacts with the protein ATP-binding cassette transporter A 1 (ABCA1) in hepatocytes [83,84] and is then secreted into circulation as a lipid-poor particle. The lipid-poor apoA-1 particle removes cholesterol from the surfaces of macrophages in the arterial wall through interaction with ABCA1, forming nascent prebeta HDL or HDL_2_ particles [85]. Phospholipid transfer protein (PLTP) is responsible for the transfer of phospholipids from triglyceride-rich lipoproteins to form nascent HDL particles and has been shown to interact with apoA-1 [86]. Apo A-1 oxidation has been shown to increase HDL oxidation and dysfunction [84,87].

The plasma enzyme [88] lecithin-cholesterol acyl transferase (LCAT) is responsible for the conversion of free cholesterol into cholesterol esters, which is a more hydrophobic form of cholesterol, making it easier to be sequestered into the core of the lipoprotein particle [89]. This eventually causes the newly synthesized HDL to assume a spherical shape [90]; the nascent HDL swells into a round, “mature” HDL particle. The mature HDL particle increases in size with the addition of more cholesterol and phospholipids from cells and other lipoproteins while circulating through the bloodstream [91].

A protein involved in the oxidation of HDL is myeloperoxidase (MPO), found in neutrophils and monocytes; it is released during acute inflammation. MPO generates Ox-HDLs, which cannot bind scavenger receptor class B type 1 (SR-B1) on the membrane of liver cells for RTC and have a proinflammatory function expressed through the upregulation of the protein (VCAM-1) on endothelial cells and the activation of NF-kB and Ox-HDL, which lose their ability to activate eNOS and to inhibit caspase-3 and, therefore, lose their antiapoptotic activity [92] (Figure 1). HDL in patients with documented heart disease have elevated levels of nitrotyrosine and chlorotyrosine, both products of myeloperoxidase oxidative processes that facilitate the generation of dysfunctional HDL [93].

Alpha-HDL [72], containing free cholesterol and a cholesterol ester core, returns to the liver for selective uptake of cholesterol via contact with SR-B1. Cholesterol ester transfer protein (CETP) regulates the exchange of cholesterol esters from HDL to apo-B-containing lipoproteins for triglycerides [94,95] Apo-containing lipoproteins then transfer cholesterol to the liver through interaction with hepatocyte LDL receptors [96].

The above processes are controlled by additional regulatory input from other proteins, including hormones and receptors. For example, the peroxisomal proliferator-activated receptor-alpha (PPARα) has been shown to upregulate transcription of the ApoA-I gene [97]; transcription of the gene for ABCA1 is regulated by liver X receptors (LXRs) and retinoid X receptors (RXRs) [98]. Each of the HDL proteins, RCT pathway proteins, and regulatory factors highlighted above represent a potential point of intervention to raise the HDL-C number and/or promote reverse cholesterol transport.

## 4. Sexual Dimorphism and HDL

There are major differences in HDL levels between men and premenopausal women [99]. Obesity affects cardiometabolic function in both men and women but affects premenopausal women to a much lesser degree, even when matched to age and weight controls [100]. Sexual dimorphism is important to understand since there are major sex differences in fat distribution in visceral organs, skeletal muscle, and epicardial fat [101] Adipose tissue distribution and adipose tissue health are responsible for differences in insulin sensitivity and consequent systemic inflammation. HDL levels are much higher in premenopausal women, but our group has shown a much higher Ox-HDL/HDL ratio in obese women, reducing the anti-inflammatory index of HDL and increasing their risk of endothelial cell dysfunction.

## 5. Cardiorenal Syndrome: A Definition

Cardiorenal syndrome is an umbrella term that encompasses the interaction between the heart and kidneys, such that injury to one organ causes dysfunction in the other [102]. There is well-established crosstalk in the pathophysiology of the heart and kidneys [103,104,105,106]. Patients on dialysis with end-stage renal disease (ESRD) have ten times the risk of death by a cardiovascular event than the general population [107]. Similarly, patients admitted with heart failure acquire renal dysfunction, and combined heart and kidney failure is associated with poor clinical outcomes [108]. The primary dysfunctional organ can be the heart or the kidney [109]. Cardiorenal syndrome is now defined more strictly than epidemiological outcomes in heart failure and dialysis patients. There are currently five types of cardiorenal syndrome, each with unique pathophysiology and progression of illness [110,111] (Table 1).

Type 1 cardiorenal syndrome is acute in nature, categorized as acute heart failure that leads to AKI, clinically presenting with inadequate renal perfusion due to an increase in venous pressure or a low cardiac output state, leading to kidney congestion [112] (Figure 2). Type 1 CRS is a common occurrence; AKI occurs in about 25% of hospitalized patients with HF, and declining renal function has been identified as an independent predictor of mortality. AKI activates the renin–angiotensin–aldosterone system (RAAS), salt and water imbalance, and vasoconstriction, all of which contribute to continued heart damage [113]. HF leads to decreased renal perfusion, along with monocyte and endothelial activation, causing cytokine secretion and further depressing renal function. This bidirectional pathophysiology is also exacerbated with common therapeutic agents such as ACE inhibitors and diuretics, leading to toxicity and vasoconstriction [114]. Production of epinephrine and angiotensin, along with a decreased sensitivity to vasodilators such as nitric oxide, causes excessive vasoconstriction, thereby worsening cardiorenal function. Fibrosis from inflammation is a common feature in HF and CKD [115] and may well be the unifying pathophysiology of the entire CRS continuum [116]. Evaluation of creatinine levels can be misleading, as the SOLVD trial showed that early initiation of RAAS inhibitors may reflect early changes in renal hemodynamics and may not reflect kidney injury [117,118]. Signs of CRS include an increase in the serum creatinine by 0.3 mg/dl in a 48-h period, an increase in the serum creatinine to 1.5 times baseline, or a urine volume less than 0.5ml/kg/h over a six-hour period [119,120].

Type 2 CRS is similar to type 1; however, where type 1 is acute, type 2 CRS is chronic (Figure 3). Type 2 CRS is characterized by chronic cardiac dysfunction (e.g., chronic congestive heart failure), which causes progressive CKD. The prevalence of renal dysfunction in chronic HF is significantly large; nearly 50% of patients with chronic HF appear to have decreased GFR. Even a minimal decrease in GFR is a strong independent predictor of mortality. Neurohormonal activation is also present in type 2 CRS, namely, increased production of vasoconstrictive mediators (angiotensin) and altered release of vasodilatory mediators (nitric oxide) [121]. Progression of CKD can be attributed to multiple factors including, but not limited to, low cardiac output, inflammation, endothelial dysfunction, accelerated atherosclerosis, chronic hypoperfusion, and increased renal vascular resistance. The progression of CKD further instigates cardiac dysfunction through RAAS activation, hypertension, and anemia.

Type 3 CRS is also called renocardiac syndrome and is categorized with abrupt kidney injury being the primary illness (e.g., ischemia, hypoperfusion, glomerulonephritis), leading to acute cardiac dysfunction (e.g., HF, arrhythmia, ischemia; Figure 4). AKI is a powerful predictor of hospital mortality; however, type 3 CRS is less common than type 1 CRS. AKI negatively impacts cardiac function through a variety of mechanisms. Renal ischemia has been shown to induce inflammation and apoptosis in cardiac cells. AKI contributes to acute heart dysfunction via familiar mechanisms from the abovementioned CRS types: RAAS activation, hypertension, decreased GFR, endothelial activation, and cytokine secretion [113]. Cytokines such as tumor necrosis factor (TNF), IL-1, and IL-6 play a diagnostic role and are also a pathogenic cause of myocardial cell damage and apoptosis during ischemic AKI. Furthermore, myeloperoxidase, a biomarker of oxidative stress and inflammation in acute coronary syndrome, may cause apoptosis and play a potential role in the pathogenesis of CRS.

Type 4 CRS is also known as chronic renocardiac syndrome, which involves primary CKD that contributes to worsening heart function and an increased risk for cardiovascular events (Figure 5). Current estimates for CKD in the general population exceed 10% of the US adult population. CKD is an independent risk factor for cardiovascular-event-related mortality in individuals at all stages of CKD, especially for ESRD, which has an increased risk of cardiac death compared to patients without CKD. CKD is considered a more significant predictor of cardiovascular disease than diabetes mellitus [113]. Mechanisms for cardiac remodeling and decreased function from CKD are multifarious in nature. In Stages 1 and 2, risk factors resulting in CKD (e.g., obesity, hypertension, dyslipidemia, and chronic inflammation) contribute to decreased cardiac function. In Stages 3 and 4, anemia, uremia toxins, nutritional status, and BMI, along with chronic inflammation, lead to increased ischemic risk, coronary calcification, and neurohormonal abnormalities, as previously discussed. In late-stage CKD leading into dialysis, we see chronic inflammation (again), renal toxicity, endothelial dysfunction, oxidative stress, and accelerated atherosclerosis as contributing factors to the vicious cycle of cardiac dysfunction with renal failure.

Type 5 cardiorenal syndrome results in simultaneous renal and cardiac failure due to acute or chronic systemic disorders. Diabetes, sepsis, amyloidosis, and sarcoidosis are examples of such diseases affecting combined renal and cardiac function (Figure 6). Sepsis accounts for the most common condition that can acutely affect both organs, with the mechanisms being unclear but possibly involving TNF. Patients presenting with sepsis have multiorgan dysfunction over 50% of the time, especially involving cardiac and renal function, explaining the high mortality involved. [122] We observed this during the COVID-19 pandemic with systemic vascular inflammation [123,124,125,126,127].

## 6. Cardiorenal Syndrome and HO-1

Physiological interaction between organs is necessary for the optimal equilibrium and functioning of the organism. Derangements in these interactions can initiate multiorgan dysfunction. In particular, heart and kidney functions are closely interrelated through a variety of dynamic and bidirectional mechanisms [128,129]; a pathological alteration in one organ can unfavorably affect function in another distant organ.

CRS involves complex interactions at the molecular level that induce vessel inflammation, atherosclerosis, cardiac fibrosis, and hypertrophy; in addition, structural and biochemical abnormalities can adversely affect cardiovascular or renal function [130].

Bright initially deliberated the causal association between chronic kidney disease (CKD) and cardiovascular risk in 1836. Patients with CKD are among the highest risk groups for adverse cardiovascular events and cardiovascular-related mortality and, therefore, require particular clinical attention. A recent study provided insight into the pathogenesis of CRS type 1, emphasizing the pivotal role of oxidative stress in CRS type 1 [128,131]. The study revealed that levels of oxidative stress markers (myeloperoxidase, nitric oxide, copper/zinc superoxide dismutase, and endogenous peroxidase activity) were significantly higher in CRS type 1 than in acute heart failure without CRS type 1 and in healthy controls [128,131]. In particular, CRS type 1 patients presented a significant increase in circulating ROS and RNS and an increased expression of the inflammatory cytokine IL-6 [132]. Monu et al. showed that ang-II-mediated recruitment of T-lymphocytes and increased oxidative stress is decreased by the upregulation of HO-1 in a model of postischemic heart failure [56]. The results showed that HO-1 induction decreased renal vasoconstriction and fibrosis and improved renal function in both immunocompetent and T-lymphocyte-suppressed mice [133]. Interestingly, treatment with SnMP, a known HO activity inhibitor, reversed the beneficial effects of HO-1 induction, suggesting that increased levels of HO activity play a central role in preventing MI- induced cardiac and renal damage in this CRS animal model. HO-1 induction reduces postischemic pathological cardiac remodeling and, in mice with advanced heart failure and CRS, improves cardiac function and renal vasoconstriction. This renal vasoconstriction was demonstrated in a murine model of type 1 CRS, secondary to postischemic changes of LAD ligation [134].

Accumulating evidence suggests that hyperuricemia is one of the important factors that may significantly contribute to the development and progression of CRS. Elevated levels of uric acid have been associated with inflammation, oxidative stress, insulin resistance, dysglycemia, endothelial dysfunction, vascular, renal and cardiac stiffness, cardiac diastolic dysfunction, renal hyperfiltration, and proteinuria, all of which are components of CRS [135,136,137]. The significance of a westernized diet, which is high in fructose, and hyperuricemia in the development of CRS is underscored by the relationship between increased consumption of sugar-sweetened beverages, hyperuricemia, and all components of this syndrome [138]. Sodhi et al. showed that induction of HO-1 reduced expression of xanthine oxidase and NADPH oxidase, enzymatic systems that are important for ROS production, in adipocytes treated with fructose, a fuel source that increases uric acid levels [139]. Additionally, Khitan et al. showed that mice following a fructose diet presented an increase in isoprostanes that was associated with a decrease in HO-1 expression and an increase in heme levels. Isoprostanes and heme are regarded as valid markers of oxidative stress [140,141,142]. Upregulation of HO-1 presents cardio- and reno-protective functions mediated by its antioxidative, anti-inflammatory, and antiapoptotic properties. In animal models of myocardial ischemia (MI), both overexpression and pharmacological induction of HO-1 reduce infarct size and ventricular remodeling after ischemia-reperfusion damage by improving cardiac metabolism [143]. Increased HO-1 expression has a protective effect against ischemia-reperfusion injury in the kidney [144] and can correct blood pressure elevation following ang-II exposure [145].

Hyperglycemia-induced mitochondrial oxidative stress, a cause of metabolic CRS, is a contributory factor to increased risk of cardiovascular disease, which can induce cellular injury and cell dysfunction [146,147]. The molecular mechanism of mitochondrial dysfunction in CRS is driven via abnormalities involving the transcriptional coactivator peroxisome proliferator-activated receptor gamma coactivator 1α (PGC-1α), which controls the biogenesis of mitochondria and mitochondrial function in a variety of tissue and cell types. PGC-1α is regulated by the endothelial NO synthase that plays an important role in mitochondrial biogenesis. Studies have shown that decreased expression of PGC-1α-associated impairment of mitochondrial biogenesis may be responsible for various metabolic abnormalities in CRS [146] (Figure 7). Thus, the impairment of the complex steps in the regulation of mitochondrial biogenesis may contribute to CRS.

Activation of PGC-1α reduced mitochondrial ROS, prevented adipogenesis in adipocytes, and protected diabetic hearts from hyperglycemia-mediated oxidative stress [148,149,150]. It has been shown that the protective effect of EETs in diabetic mice involves increased expression of PGC-1α and SIRT1 [131]. In cardiomyocytes, pharmacological inhibition of SIRT1 was followed by decreased expression of PGC-1α and increased ROS production [150,151]. At the same time, Singh et al. showed that PGC-1α regulates HO-1 expression, confirming the beneficial effect of HO-1 in cardiovascular diseases and lipid metabolism [152,153]. Heme oxygenase-1 upregulation of PGC-1α signaling in epicardial fat attenuated cardiovascular risk in both humans and mice [154]. Adipocyte-specific HO-1 gene therapy is very effective in reducing oxidative stress, improving both insulin resistance and vascular function in obese mice [155]. Finally, upregulation of HO-1 by icosapent ethyl, pomegranate seed oil, and black seed oil (thymoquine) have all reduced oxidative stress, improved insulin resistance, and improved mitochondrial function [156,157] (Figure 7).

More attention is being paid to the role of Ox-HDL in the chronic inflammatory state of obesity. Ox-HDL has been shown to independently upregulate the downstream activity of ang II and has been identified as part of a biomarker profile for early endothelial dysfunction in obese women without identified cardiovascular disease [158] (Figure 1). We have shown oxidized HDL to be part of a biomarker profile for cardiovascular risk in obese women [159]. PGC-1α has been shown to improve organ function by upregulating mitochondrial enzymes, improving mitochondrial function in metabolic syndrome and nonalcoholic fatty liver disease (NAFLD), and reducing Ox-HDL [146]. Most importantly, this improvement in mitochondrial integrity and function will aid in the reprogramming of white fat to beige-like fat [160]

## 7. Lifestyle Interventions, Weight Loss Medications, and Nutraceuticals

Weight loss with diet and exercise has been successful in short-term supervised programs, with weight loss in excess of 5% body weight. None have approached the magic number of 10–12% of sustained body weight loss in BMIs above 35. The PREDIMED-PLUS study was a 12-month intervention that increased weight loss and improved cardiovascular risk factors [161]. There is confusing evidence as to the effect of aerobic exercise on HDL levels and cholesterol efflux [162]. Exercise has been shown to improve psychological health, particularly in the elderly [163]. Many nutraceuticals (like curcumin, CoQ 10, folic acid, berberine, alpha lipoic acid, astaxanthin, and policosanol) have tried to improve lipid and plasma glucose levels, with positive effects on lipid profiles [164]. Weight-loss drugs have been effective in supervised programs, especially those with close telemedicine follow-up [165,166,167,168]. There is clearly a role for diet, exercise, weight loss medications, and nutraceuticals as preventive measures in obesity, metabolic syndrome, and type 2 diabetes. They are measures meant to prevent cardiorenal syndrome. We need more aggressive measures once we are confronted with cardiorenal syndrome.

## 8. Summary

Mitochondrial function is important for improving the function of the electron transport chain in both the heart and kidneys and converting white adipose tissue to beige, the “browning“ of white adipose tissue. This results in an improvement of adiponectin levels and a severe reduction in the release of inflammatory adipocytokines. HO-1 upregulation is the key to improving mitochondrial function in both organs and reducing oxidative stress and Ox-HDL, which are important components of success in treating and preventing cardiorenal syndrome.

## Figures and Tables

**Figure 1 diagnostics-10-00976-f001:**
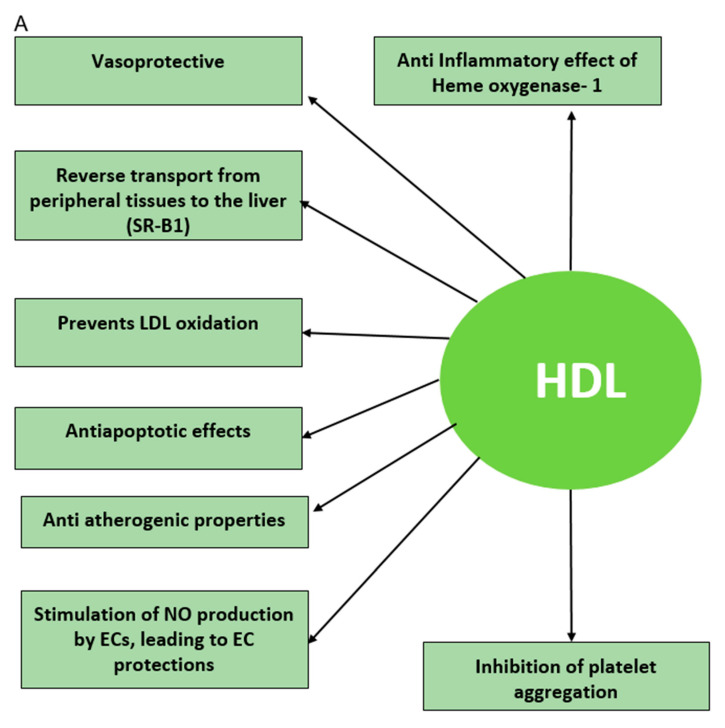
(**A**) The beneficial effects of high-density lipoprotein (HDL). HDL exhibits protective effects through multiple mechanisms, including lowering tissue cholesterol levels through reverse cholesterol transport, attenuation of LDL oxidation, and decreasing inflammatory responses. (**B**) The adverse effects of oxidized HDL (Ox-HDL). Oxidized or dysfunctional HDL is proinflammatory and atherogenic. (**C**) Role of HO-1. HO-1 has a protective effect against reactive oxygen species (ROS) and oxidative stress, and the upregulation of HO-1 decreases the detrimental effects of oxidized HDL. HO-1 = heme oxygenase 1; SR-B1 = scavenger receptor class B type 1; HDL=high-density lipoprotein cholesterol; LDL = low-density lipoprotein cholesterol; NO = nitric oxide; EC = endothelial cells.

**Figure 2 diagnostics-10-00976-f002:**
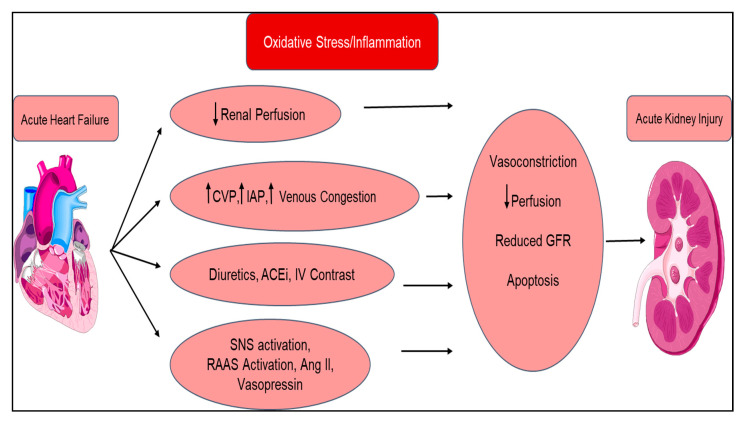
Pathophysiology of acute cardiorenal syndrome (type 1): mechanism of how acute heart failure leads to acute kidney injury (AKI) due to inadequate renal perfusion, endothelial activation, and cytokine production, which activates the RAAS, salt and water imbalance, and vasoconstriction, further exacerbating AKI. CVP = central venous pressure; IAP = intraabdominal pressure; ACEi = angiotensin converter enzyme inhibitor; RAAS = renin–angiotensin–aldosterone system; SNS = sympathetic nervous system; GFR = glomerular filtration rate.

**Figure 3 diagnostics-10-00976-f003:**
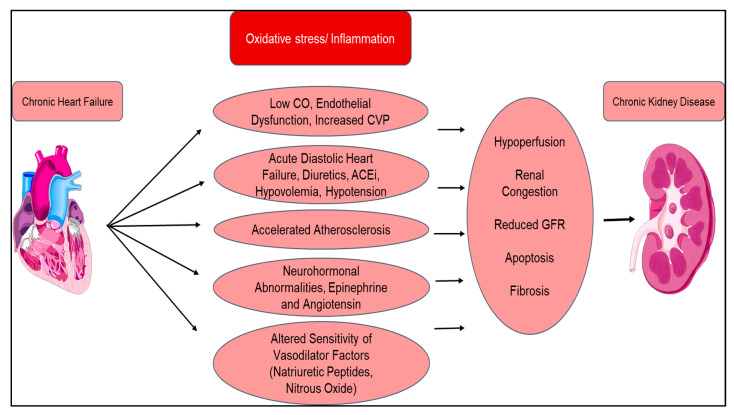
Pathophysiology of chronic cardiorenal syndrome (type 2): mechanism of how multiple effects of chronic heart failure lead to progressive and chronic kidney disease (CKD) because of neurohormonal upregulation, leading to altered vasoconstriction and vasodilation. CO = cardiac output; CVP = central venous pressure; ACEi = angiotensin-converting enzyme inhibitor.

**Figure 4 diagnostics-10-00976-f004:**
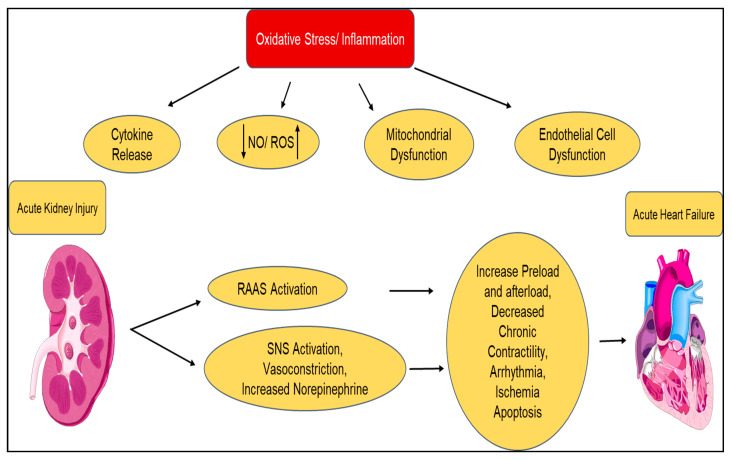
Pathophysiology of renocardiac syndrome (type 3): mechanism involves an acute insult to kidney function that results in the acute sympathetic nervous system activating a cascade of inflammatory responses, causing acute heart failure. NO = nitrous oxide; RAAS = renin–angiotensin–aldosterone system; ROS = reactive oxygen species; SNS = sympathetic nervous system.

**Figure 5 diagnostics-10-00976-f005:**
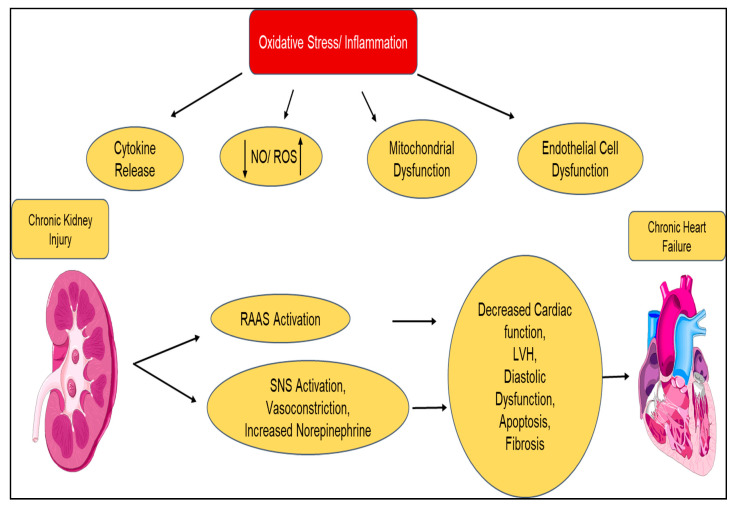
Pathophysiology of renocardiac syndrome (type 4): mechanism involving chronic kidney injury causing worsening heart function. The various causes of CKD, including diabetes and hypertension, accelerate cardiac remodeling, leading to poor cardiac function. This pathway involves RAAS activation and sympathetic nervous system activation.

**Figure 6 diagnostics-10-00976-f006:**
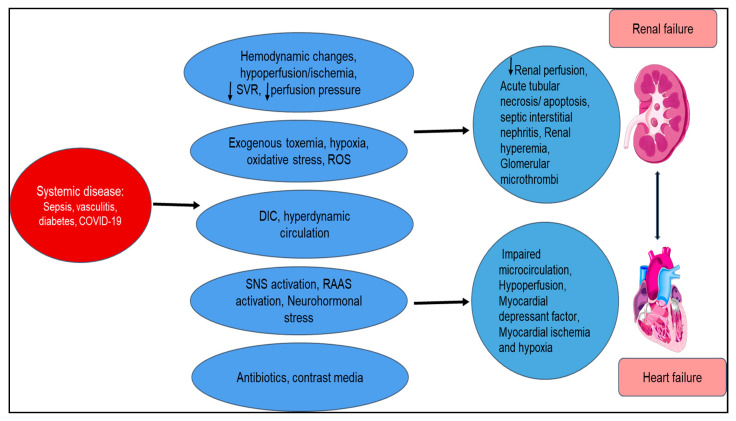
Summary of the pathophysiology of secondary cardiorenal syndrome (type 5): mechanism is characterized by combined cardiac and renal failure due to acute or chronic processes that cause hemodynamic instabilities, hypercoagulability, neurohormonal imbalances, and toxicity and hypoxia that cause poor renal perfusion, myocardial ischemia, and hypoxia. SVR = systemic vascular resistance; ROS = reactive oxygen species; DIC = disseminated intravascular coagulation; SNS = sympathetic nervous system; RAAS = renin–angiotensin–aldosterone system.

**Figure 7 diagnostics-10-00976-f007:**
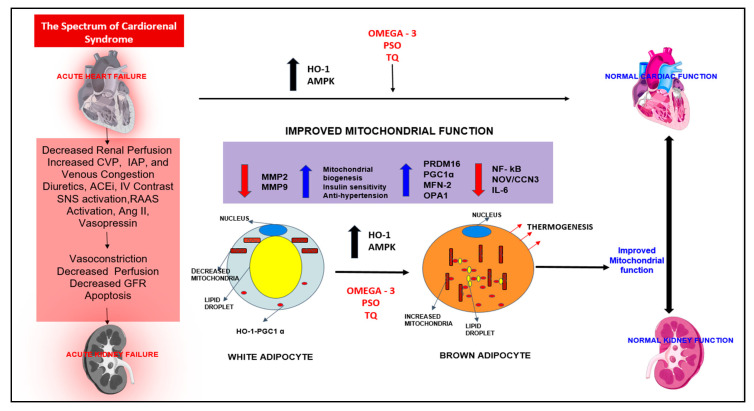
The spectrum of cardiorenal syndrome. This is a schematic representation of how HO-1 upregulation results in improved mitochondrial function and signaling pathways PGC1 α, PRDM 16, OPA1, and MFN-2. Upregulation also results in decreased inflammatory adipocytokines (NOV/CCN3, IL-6, and NF-kB) and fibrotic markers (MMP2 and MMP9). The result was the “browning” of white adipocytes to “beige”, improving cardiac and renal functions. HO-1 = heme oxygenase 1; AMPK = AMP-activated protein kinase; PSO = pomegranate seed oil; TQ = thymoquinone; NF-kB = nuclear factor kappa-light-chain-enhancer of activated B-cells; NOV/CCN3 = nephroblastoma overexpressed; IL-6 = interleukin 6; MMP2 = matrix metalloproteinase 2; MMP9 = matrix metalloproteinase 9; PRDM16 = PR domain containing 16; PGC-1 α = peroxisome proliferator-receptor gamma coactivator 1α; MFN-2 = mitofusin-2; OPA1 = mitochondrial dynamin-like GTPase.

**Table 1 diagnostics-10-00976-t001:** Types of Cardiorenal Syndrome.

**Type 1** **Acute Cardiorenal Syndrome**	Acute decompensated heart failure ⇨ Acute kidney injury
**Type 2** **Chronic Cardiorenal Syndrome**	Chronic heart failure ⇨ Chronic kidney disease
**Type 3** **Acute Renocardiac syndrome**	Acute kidney injury ⇨ Acute heart failure
**Type 4** **Chronic Renocardiac syndrome**	Chronic kidney disease ⇨ Chronic heart failure
**Type 5** **Secondary Cardiorenal syndrome**	Codevelopment of heart failure and chronic kidney disease due to acute or chronic systemic disorder

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
