# Peer review of "OX-HDL: A Starring Role in Cardiorenal Syndrome and the Effects of Heme Oxygenase-1 Intervention"

_diagnostics, 2020, doi:10.3390/diagnostics10110976_

Round 1
Reviewer 1 Report
think that this is a very interesting paper, well written and with clear summary.
There is one important point which needs to be addressed before the publication: the
Other minor comments:
- Authors show no discussion about sexual dimorphism. I think that this aspect must have an answer.
Author Response
Reviewer 1
There is one important point which needs to be addressed before the publication: the
- Authors show no discussion about sexual dimorphism. I think that this aspect must have an answer.
- We want to thank this reviewer for his thoughtful comments, which has helped dramatically improve this paper. We put this paper through a grammar and spell check review, and have made all the necessary corrections.
- We have added a paragraph on sexual dimorphism and HDL as requested. This was an excellent suggestion and we are grateful for the advice. We thank you gain for your valuable time in helping us improve this paper.
- new paragraph added
-
4-Sexual dimorphism and HDL
There are major differences in HDL levels between men and pre-menopausal women [99] Obesity affects cardiometabolic function in both men and women, but affects pre-menopausal women to a much lesser degree, even when matched to age and weight matched controls [100] Sexual dimorphism is important to understand since there are major sex differences in fat distribution in visceral organs, skeletal muscle and epicardial fat [101]Adipose tissue distribution and adipose tissue health is responsible for differences in insulin sensitivity and consequent systemic inflammation. HDL levels are much higher in pre-menopausal women, but our group has shown a much higher Ox-HDL/HDL ratio in obese women, reducing the anti-inflammatory index of HDL and increasing their risk of endothelial cell dysfunction [156].
Reviewer 2 Report
it may be appropriate to add another paragraph before paragraph 6 (summery) on the data available in literature on the possible effects of lifestyle, diet, drug therapies or nutraceuticals on HDL levels
Author Response
Reviewer 2
Comments and Suggestions for Authors
it may be appropriate to add another paragraph before paragraph 6 (summery) on the data available in literature on the possible effects of lifestyle, diet, drug therapies or nutraceuticals on HDL levels
We want to thank this reviewer for his thoughtful comments, which has helped dramatically improve this paper. We put this paper through a grammar and spell check review, and have made all the necessary corrections. We have added the paragraph on diet, exercise, medications and nutraceuticals. This was an excellent suggestion and we are grateful for the advice. We thank you again for your valuable time in helping us improve this paper.
New paragraph added
- 7. Lifestyle interventions, Weight loss medications and Nutraceuticals
Weight loss with diet and exercise have been successful in short term supervised programs with weight loss in excess of 5% body weight. None have approached the magic number of 10-12% of sustained body weight loss in BMI’s above 35. The PREDIMED-PLUS study was a 12 month intervention that increased weight loss and improved cardiovascular risk factors [161] There is confusing evidence of the effect of aerobic exercise on HDL levels and cholesterol efflux [162] Exercise has been shown to improve psychological health, particularly in the elderly [163] Many nutraceuticals have been tried to improve lipid and plasma glucose levels, like curcumin, CoQ 10, folic acid, berberine, alpha lipoic acid, astaxanthin and policosanol, with positive effects on lipid profiles [164] Weight loss drugs have been effective in supervised programs, especially those with close telemedicine follow up [165-168] There is clearly a role for diet, exercise, weight loss medications and nutraceuticals as preventive measures in obesity, the metabolic syndrome and Type 2 diabetes. They are measures meant to prevent the cardiorenal syndrome. We need more aggressive measures once we are confronted with the cardiorenal syndrome.